# Mechanistic Insights into the Roles of the IL-17/IL-17R Families in Pancreatic Cancer

**DOI:** 10.3390/ijms241713539

**Published:** 2023-08-31

**Authors:** Zheng Chen, Shuangying Qiao, Liu Yang, Meiheng Sun, Boyue Li, Aiping Lu, Fangfei Li

**Affiliations:** 1Shum Yiu Foon Shum Bik Chuen Memorial Centre for Cancer and Inflammation Research, School of Chinese Medicine, Hong Kong Baptist University, Hong Kong SAR, China; zacharychancz@hkbu.edu.hk (Z.C.); qiaosysy@hkbu.edu.hk (S.Q.); yangliu_rachel@life.hkbu.edu.hk (L.Y.); 20481004@life.hkbu.edu.hk (M.S.); 22420487@life.hkbu.edu.hk (B.L.); 2Institute of Precision Medicine and Innovative Drug Discovery (PMID), School of Chinese Medicine, Hong Kong Baptist University, Hong Kong SAR, China

**Keywords:** IL-17, pancreatic cancer, chemotherapy resistance, tumor immune microenvironment

## Abstract

The members of the cytokine interleukin 17 (IL-17) family, along with their receptors (IL-17R), are vital players in a range of inflammatory diseases and cancer. Although generally regarded as proinflammatory, the effects they exhibit on cancer progression are a double-edged sword, with both antitumor and protumor activities being discovered. There is growing evidence that the IL-17 signaling pathways have significant impacts on the tumor microenvironment (TME), immune response, and inflammation in various types of cancer, including pancreatic cancer. However, the detailed mechanistic functions of the IL-17/IL-17R families in pancreatic cancer were rarely systematically elucidated. This review considers the role of the IL-17/IL-17R families in inflammation and tumor immunity and elaborates on the mechanistic functions and correlations of these members with pathogenesis, progression, and chemoresistance in pancreatic cancer. By summarizing the advanced findings on the role of IL-17/IL17R family members and IL-17 signaling pathways at the molecular level, cellular level, and disease level in pancreatic cancer, this review provides an in-depth discussion on the potential of IL-17/IL-17R as prognostic markers and therapeutic targets in pancreatic cancer.

## 1. Introduction

Pancreatic adenocarcinoma (PAAD) is a highly malignant and lethal tumor. The most common type of pancreatic neoplasm is pancreatic ductal adenocarcinoma (PDAC), which accounts for over 90% of pancreatic cancer cases. Due to the difficulty of early detection, the 5-year survival rate for pancreatic cancer patients remains below 10%, with a median survival time of only 2–3 months [1]. Moreover, patients with pancreatic neoplasm have a poorer prognosis than those with other malignancies [2,3]. Over the last decade, the incidence and mortality of pancreatic cancer have significantly increased. From 2009 to 2019, the number of deaths due to pancreatic cancer has risen by 65% in Hong Kong [4]. Despite numerous cutting-edge research studies discussing chemotherapy resistance and targets or receptors, the gateway to overcoming pancreatic cancer remains indispensable. Thus, it is crucial to explore novel targets, advanced mechanisms, and pathways to enhance the efficiency of chemotherapy and improve the clinical outcomes of pancreatic cancer patients.

The members of the cytokine interleukin-17 (IL-17) family and their receptors (IL-17R) are indispensable contributors to various inflammatory conditions and malignancies. IL-17 secretion is primarily produced in T-helper-17 (Th17) cells, which are a subset of CD4+ T cells [5]. Apart from Th17 cells, IL-17 is also generated by specific subsets of cells, such as γδ T cells and natural killer cells [6]. The tumor microenvironment regulated by Th17 cells and other immune cells may be involved in tumor progression by interacting with different interleukins and tumor cells [7,8]. Previous studies have shown that IL-17/IL-17R are involved in the progression of inflammatory ailments and immune responses and also play a crucial role in the tumor microenvironment of PDAC [9]. The abundance of IL-17 is significantly higher in PDAC compared to adjacent tissues based on the pathway and mechanism of IL-17 discovered in different types of malignant tumors [10]. Chemotherapy is currently the most commonly used first-line treatment for pancreatic cancer patients, but the biological mechanisms and biomarkers of chemotherapy resistance are still unclear [11]. Several pathways, such as IL-17, MAPK, and NF-κB, have been identified as modulators of toxicity and drug resistance [7,12]. Given the known functions of IL-17 in pancreatic cancer, investigating the role of the IL-17/IL-17R pathway in the development and progression of pancreatic cancer could be a promising approach for improving treatment outcomes.

Therefore, in this review, we will first introduce the family and classification of IL-17/IL-17R. Second, we will highlight the important biological features of IL-17/IL-17R in tumor immunity and inflammation and provide an overview of its pathway. Third, we will reveal the crucial role of IL-17/IL-17R in chemotherapy resistance within the pancreatic tumor immune microenvironment. In this review, we summarize the relevant literature reviews on IL-17 and IL-17R in recent years. Since IL-17 was first discovered as an inflammatory cytokine that affects autoimmune diseases such as psoriasis, here, we summarize the signaling mechanism of IL-17 in the pathogenesis of pancreatic cancer, which may provide new opportunities for therapeutic intervention. Investigating the IL-17/IL-17R and its pathway could lead to the development of novel strategies to overcome clinical obstacles in pancreatic cancer, which could potentially improve patient outcomes.

## 2. The Overview of the IL-17/IL17R Families

### 2.1. The IL-17 Family Members

The interleukin 17 family is composed of six different ligands, ranging from IL-17A to IL-17F (as shown in Figure 1). Interleukin-17A (IL-17A), which was originally known as CTLA-8, was first identified in 1993 through mouse T-cell library cloning [13]. It is a characteristic marker of the IL-17 family in pancreatic diseases [10]. Recent studies have proposed that IL-17A could act as a potential treatment option for pancreatic cancer by crossing a mouse model of idiopathic PDAC (KPC) with IL-17A knockout mice and that cells with intact IL-17A and IL-17A-deficient cells had very different characteristics [14]. These differences were attributed to the elevated serum levels of IL-17F in IL-17A (−/−) mice and the high expression of its specific cognate receptor (IL-17RC) in IL-17A (−/−) tumor-associated fibroblasts (CAFs) [14]. Combining gemcitabine treatment with an anti-IL17A antibody could be an effective strategy to regulate the function and activity, such as the metabolism of macrophages, by attenuating typical M1 and M2 metabolic pathways and elicit antitumor responses, particularly in cases of pancreatic cancer where gemcitabine is frequently used [15]. These results indicate that targeting the IL-17A pathway could enhance the effectiveness of gemcitabine and potentially improve treatment outcomes for patients with pancreatic cancer.

IL-17B is widely expressed in the stomach, pancreas, intestine, and other tissues, and it affects the pathogenesis of malignant tumors such as gastric and pancreatic cancer [16]. In mice, IL-17B was found to increase during intestinal inflammation in the abdominal cavity and accelerated the migration of neutrophils [13]. Moreover, a study revealed that IL-17B levels are highly expressed and associated with the prognosis of pancreatic, lung, and breast cancers [16,17,18,19]. It has a similar effect to IL-17A, promoting the invasion and metastasis of pancreatic cancer and predicting both the prognosis of pancreatic cancer and the efficacy of gemcitabine treatment [18]. These findings suggest that targeting the IL-17B could also provide a promising direction for the treatment of pancreatic cancer.

IL-17C is strongly expressed in the mucosa of the mouth, skin, and airway epithelium [20]. It can respond to other cytokines and pathogenic stimuli at mucosal surfaces and play a role in autoimmune, autoinflammatory, and bacterial diseases [20]. When the herpes simplex virus-2 reactivates, the neurotrophic cytokine IL-17C is adequate to support nerve growth and protect peripheral neurons [21]. IL-17C has also been linked to a pathologic microbiota that is present in patients with chronic obstructive pulmonary disease and may promote tumor growth [22]. Further research is needed to explore the potential of IL-17C as a therapeutic target for inflammation-related conditions.

IL-17D, on the other hand, is poorly documented and perhaps the least understood member of the IL-17 family. It is expressed in many healthy tissues, and its expression is increased in the presence of tumors or viral infections [23]. IL-17D is abundantly and differently expressed in tumors and is at low levels in poorly immunogenic and edited tumors but highly expressed in immunogenic and unedited tumors [24].

IL-17E (also called IL-25) has a strong inflammatory effect both in vivo and in vitro [25]. An increased expression of IL-17E leads to a T-helper-2 (Th2)-type immune response in mice that triggers an increase in the expression of IL-4 and IL-13 in multiple tissues, as well as the expansion of eosinophils by way of the release of IL-5 [26].

The last chemokine of the family, IL-17F, is a crucial regulator of inflammatory response and cancer development [27]. Although IL-17F and IL-17A share 50% homology, they play different roles in immunity, inflammation, and tumorigenesis. The IL-17A/IL-17F axis promotes the proliferation and progression of lung cells by regulating macrophages [28].

IL-17A and IL-17B are primarily involved in the regulation of the tumor immune microenvironment, and they particularly play a special role in the drug resistance and inflammatory response in pancreatic cancer, according to our overview of six different cytokines. In the content that follows, the specific signaling pathway mechanism will be covered. In addition, it has been discovered that a number of additional IL-17 ligand family members are expressed in tumor tissues and are also crucial for the growth and development of tumors.

### 2.2. The IL-17 Receptor Family Members

IL-17 also consists of five different receptor numbers, including IL-17 receptor A (IL-17RA), B, C, D, and E (Figure 1). IL-17RA was discovered in 1995 as a cytokine receptor for IL-17A, belonging to a family distinct from existing cytokine receptors [13]. Later, IL-17RB, RC, RD, and RE were discovered, all of which bind to their ligands and share sequence homology [29]. IL-17RA was initially connected with its ligand IL-17A but was later found to also bind to IL-17F [30]. However, the composition of complexes containing IL-17RA is not yet clear. Independent complexes containing IL-17RA have been discovered through fluorescence resonance energy transfer (FRET) [31]. In addition to directly engaging IL-17 cytokines, IL-17RA may also function as a co-receptor within IL-17 cytokine complexes [29]. It is involved in other pathways such as the signal transducer and activator of transcription (STAT), NF-κB, Activator protein 1 (AP-1), Mitogen-activated protein kinases (MAPK), mRNA stability, and CCAAT-enhancer-binding proteins (C/EBPs) [32].

IL-17RB (also called IL-25R, Evi-27, or IL-17rh-l) binds its ligands IL-17B and IL-17E/IL-25. It is expressed in various organs and endocrine tissues, including the kidney, liver, and mucosa [16]. Patients with asthmatic or autoimmune diseases consistently have elevated expression of IL-17RB in their lung tissues [16]. IL-17RB also plays a unique role in contributing to tumor development and invasion and migration upon stimulation with IL-17B. IL-17B signaling directly promotes the proliferation and migration of cancer cells through IL-17RB and induces resistance to traditional chemotherapy drugs [33]. IL-17RC (also known as IL-17RL) also plays a central functional role in regulating IL-17 reactions, especially interacting with IL-17A and IL-17F [34]. Unlike IL-17RA, prostate, liver, kidney, thyroid, and joint cells express low levels of IL-17RC compared with hematopoietic tissues [29].

Interleukin-17 receptor D (IL-17RD) is also affiliated with the IL-17 receptor family and has evolved gradually. Initially discovered to regulate FGF signaling as an inhibitor of fibroblasts, IL-17RD also regulates other receptor tyrosine kinase signaling pathways [35]. Recent studies have shown that IL-17RD has a connection with IL-17 and Toll-like receptors (TLRs), and genetic and cell biology studies have suggested that it controls cell proliferation, differentiation, and inflammation [35]. There is also growing evidence that IL-17RD is involved in tumorigenesis, as the downregulation of IL-17RD has been observed in various human cancers, and the loss of IL-17RD promotes tumor formation in mice [36]. Compared with other receptors in the IL-17 family, fewer studies have been conducted on IL-17RE. However, the pathway in inflammation and immunity between IL-17RE and IL-17C has been suggested, and high IL-17 and IL-17RE expression has been linked to poor outcomes in HCC patients [37].

We introduced five different receptors of IL-17 separately. It is obvious that the three receptors IL-17RA/RB/RD play a key role in affecting the signaling pathway of tumor progression and tumor development. Next, we will continue to reveal the specific mechanism and role of these key receptors in pancreatic cancer. At the same time, the exploration of IL-17RC and IL-17RE cannot be stopped. Since they belong to the same receptor family, subsequent research on these two receptors and tumor development may also make some progress.

There are five classical receptors, and their corresponding ligands are displayed in Figure 1. Obviously, IL-17 and IL-17F exist either as homodimers or as a heterodimer to bind the corresponding receptor complex, in which IL-17RA acts as the most common subunit for all the other receptors [38]. In addition, these receptor complexes share a common cytoplasmic motif known as the SEF-IL-17R (SEFIR) motif, adjacent to a SEFIR extension (SEFEX) domain, and also consists of an inhibitory CBAD domain [39]. Act1 is an essential component in the induction of inflammatory genes and acts as a membrane-proximal adaptor of the IL-17 receptor [40]. The essential adaptor molecule ACT1 is recruited to the IL-17R directly and is necessary for the transcriptional and posttranscriptional changes triggered by IL-17 [41]. Overall, we summarize all the members of the IL-17 family and their receptors in Table 1, and understanding the specific mechanism and how each of these factors interacts with the others to influence IL-17-induced signaling pathways in inflammation and tumor immunity are crucial.

### 2.3. The IL-17 Signaling Pathways

To gain a better understanding of the IL-17/IL-17R pathway, it is crucial to investigate the signal transduction of IL-17. A wide variety of signaling activators, such as IFN-γ, IL-13, TGF, and even microbial compounds, are effectively coupled with IL-17 [49,50]. IL-17A performs a unique profibrotic role by improving hepatic stellate cells’ reactivity to TGF-β via activation of the JNK pathway. TGF-RII is upregulated and stabilized by IL-17A, which increases SMAD2/3 signaling [51]. Moreover, the lung pathology and the upregulation of IL-13-driven transcripts are prompted by IL-17A and IL-13-induced STAT6 activation [52]. Several studies have demonstrated that IL-17A activates nuclear factor-κB (NF-κB) and ACT1, which plays an important role in inducing inflammation by acting as a membrane-proximal adaptor for the IL-17 receptor [40]. In liver cancer cell lines, IL-17A may stimulate angiogenesis induced by CXC chemokines, independently of VEGF signaling, to promote tumor progression in mice [53]. IL-17A activates multiple MAPKs, with ERK being phosphorylated more intensely and faster [29]. IL-17RA signaling requires TRAF6, which suggests that it might be similar to TLR/IL1R signaling [54]. Another study found that IL-17-induced IL-6 production from the TEM triggers tumor-intrinsic STAT3, promoting tumorigenesis [55]. Meanwhile, copper uptake is induced when the IL-17, STAP4, and XIAP axes are activated by the inflammatory response, leading to the production of colon cancer [56]. In colorectal cancer, increasing miR-146a expression or inhibiting miR-146a target expression may be a therapeutic strategy that limits the pathways leading to tumorigenic IL-17 signaling [57]. In ovarian cancer, MTA1 mRNA and protein expression are increased by IL-17, which promotes the migration and invasion [58].

In a recent study, breast cancer patients with high levels of IL-17RB had a higher correlation with poor prognosis compared to patients with high levels of HER2 [19]. The study also suggests that IL-17RB promotes the progression of breast cancer through the NF-κB pathway [19]. IL-17B/IL-17RB signaling and ERK are involved in a significant pathway that enhances the invasion and migration of cancer cells [17]. In gastric cancer, the proliferation and migration of tumor cells can be promoted via the IL-17B/IL-17RB axis, which also upregulates cell stemness by triggering the AKT/β-catenin pathway [59]. Therefore, IL-17RB may be a significant marker in therapy for gastric cancer patients. In clinical cases where Th17 cells’ responses are necessary for disease pathogenesis, notch signaling inhibitors may benefit patients in vivo by suppressing the Th17 cell response [60]. The blockade of IL-17 signaling has been shown to increase the expression of immune checkpoint markers. For instance, the treatment of mice with colorectal cancer with anti-CTLA-4 antibody has increased the expression of protumor IL-17 [61]. To ensure effective treatment, it is important to understand which pathway of the receptor to target without adversely affecting the patient’s immune system.

### 2.4. The Role of IL-17/IL17R Families in Inflammation and Tumor Immunity

As a proinflammatory cytokine, the member of the IL-17 family plays a crucial role in the development of inflammatory diseases and the pathogenesis of several types of inflammation. Recent studies suggest that IL-17 also contributes to tumor progression, chemotherapy resistance, and tumor development through interactions with other inflammatory cytokines [62]. IL-17/IL17R complex triggers inflammatory responses by inducing proinflammatory cytokines and chemokines from epithelial and stromal cells. The activation of IL-17 also creates a favorable environment for tumor growth, which depends on the different IL-17 family numbers to increase inflammatory mediators, mobilize immune cells, and alter the phenotype of stromal cells [41]. The IL-17 family induces molecules such as chemokines, cytokines, anti-microbial peptides, and matrix metalloproteinases [63]. It also triggers the induction of several mediators that promote inflammation, including G-CSF, IL-6, and CXCL1; recruit immune cells; and establish an immune suppressive tumor environment [64]. Additionally, IL-17 has been shown to promote prostate carcinogenesis by inducing EMT through the mediating effects of MMP7. This suggests that the IL-17-MMP7-EMT axis pathway may provide a direction for exploring new strategies for the prevention and treatment of prostate cancer in the future [65]. IL-17A has been shown to stimulate the secretion of angiogenic CXC chemokines, including CXCL1, in live cancer cells, leading to faster growth of implanted syngeneic tumors [66]. In DMBA/TPA-induced skin carcinogenesis, IL-17 is essential for inflammation-associated tumor growth, and IL-17 generates tumor-promoting inflammation through a variety of interacting pathways [67]. IL-17B and IL-17C have been found to work together with IL-17A and IL-17F to increase proinflammatory activity by regulating the release of TNF-α and IL-1β [68]. This cooperation can enhance the effects of IL-17A and IL-17F on the local secretion of C-X-C chemokines and growth factors in tissues [68]. These findings show that inflammation mediated by IL-17 is a critical pathway for the inflammation-mediated promotion of growing tumors.

Th17 cells are a specific type of CD4^+^ helper T cell that release IL-17. These cells have been linked to inflammation in tumors, promoting the migration and invasion of cancer cells, as well as increasing the production of angiogenic factors [69]. Interestingly, IL-17 can also be expressed by other immune cells, including natural killer T (NKT) cells, innate lymphoid cells, CD8^+^ T cells, and non-CD8^+^ T cells [8]. While Th17 cells are known to play a crucial role in inflammation and autoimmune disorders, their significance in the tumor microenvironment remains uncertain. Recent research has revealed the existence of a non-canonical CD8+ T-cell subpopulation that produces IL-17A, which can promote pancreatic cancer progression through IL-17RA signaling [70]. Several studies have explored the functions of IL-17 in tumor immunity, which represent both antitumorigenic and pro-tumorigenic effects [71]. There are certainly different dynamics of Th17 cells transforming in the microenvironment to shape the effect of Th17 cells on different cancers. On the one hand, TGF-β facilitates an alteration in differentiation into the Treg phenotype, while IL-23 and IL-6 maintain the Th17 phenotype; on the other hand, in the case of a deficiency of TGF-β, IL-23 and IL-12 induce the conversion of Th17 cells to a Th1 cell phenotype [72,73]. The molecular pathways of PGE2 released by breast tumor cells in increasing IL-23 production and Th17 cell proliferation are highlighted in the study [74]. IL-17 is primarily secreted by Th17 lymphocytes and can impact carcinogenesis, cancer cell proliferation, and metastasis. For example, IL-17-producing γδ T cells and neutrophils may promote breast cancer metastasis [75], and IL-17 has been shown to promote tumorigenesis in renal cell carcinoma [76]. Meanwhile, the high expression of the IL-17B/IL-17RB axis has been associated with poor prognosis of several cancers, including breast cancer, lung cancer, and gastric cancer [17,19,59]. While IL-17 has the potential to act as an oncogene in tumorigenesis and metastasis, there is also evidence that it can act as a tumor suppressor during these processes. An immune response mediated by IL-17-positive cells and T-regs has been shown to be beneficial in cervical adenocarcinoma, while a response mediated by Th17 cells may be detrimental [77]. Therefore, the local immune response in cervical adenocarcinoma may contribute differently to tumor growth compared to other cancer types.

Immune cells dominate the tumor microenvironment, and Th17 cells have received significant attention in recent years due to their distribution and function in the immune response. The immune escape mechanism of cancer involves many factors, with the programmed cell death protein being a prominent one. IL-17 has been shown to upregulate PD-L1 expression by stimulating the expression of NF-κB and ERK1/2 signaling in human prostate and colon cancer cells [78]. However, it is still unclear whether IL-17 and TNF-α act synergistically or individually to induce PD-L1 expression. Studies have shown that the induction of Th17 cells in the tumor microenvironment can inhibit tumor growth and improve survival rates, suggesting that Th17 cells may have antitumor effects. Th17 cells and their related cytokines IL-12, IL-17, and IL-23 may also affect the prognosis of pancreatic cancer patients [79]. In a murine model of pancreatic cancer, IL-6 alters the balance of Th17 cells in the tumor microenvironment and the induction of Th17 cells in the tumor microenvironment to inhibit tumor growth [80]. However, the role of Th17 cells and IL-17 in cancer is complex and context-dependent. It is necessary to fully understand their function in inflammation and tumor immunity and establish new guidelines for cancer treatment in further research. Targeting IL-17 family members could therefore be a valuable approach to predicting cancer prognosis and guiding immunotherapy in cancer treatment.

## 3. Mechanistic Functions of the IL-17/IL17R Families in Pancreatic Cancer

### 3.1. The Role of IL-17/IL17R Families in the Pathogenesis of Pancreatic Cancer (PanIN and ADM Stages)

Pancreatic ductal adenocarcinoma (PDAC) is a devastating disease with a poor prognosis that is often diagnosed at an advanced stage. Therefore, it is crucial to have a comprehensive understanding of its early occurrence and transformation process to develop diagnostic markers and early interventions. In the presence of oncogenic KRAS mutations and acute or chronic inflammation, mature pancreatic acinar cells have a high degree of plasticity and can differentiate into ductal-like cells with ductal cell traits, a process known as acinar-to-ductal metaplasia (ADM). This transformation can trigger the progression to pancreatic intraepithelial neoplasia (PanIN), a precursor lesion of PDAC [81,82]. PDAC arises from a variety of precursor lesions, including PanIN and intraductal papillary mucinous neoplasms (IPMN). Based on reports, IL-17 provides an essential part in the initial development and progression of PDAC pancreatic precursor lesions [83]. It has been reported that IL-17A accelerates pancreatic acinar–ductal metaplasia, contributes to the maintenance of stem-like cancer cells, and recruits immunosuppressive granulocytes to the tumor site [14,84]. The IL-17 axis and Notch pathway have been shown to have a synergistic effect in promoting the development of PanIN and PDAC [85] (Figure 2B). The activation of an IL-17 signaling axis between hematopoiesis and epithelium is one of the key factors driving the formation of PanIN, while the inhibition of IL-17 signaling using pharmacologic strategies effectively decreased PanIN formation [84]. Additionally, immune-cell-derived IL-17 can regulate the stem cell features of pancreatic cancer cells, leading to the increased expression of DCLK1, POU2F3, ALDH1A1, and IL-17RC [86]. Moreover, a previous study found that IL-17 induces the expression of REG3β, which is recognized as a mediator of pancreatitis involved in early PanIN lesions and acinar-to-ductal metaplasia [87]. As we have mentioned, Th17 cells associate with proinflammatory cytokines and regulated immune responses in PDAC patients, and the FOXP3+RORγt+Treg has been found to interact with Th17 cells, which may be involved in the pathogenesis of PDAC by regulating the activity of proinflammatory and immunosuppressive [88]. These findings highlight the intricate interplay between immune cells and the pancreatic epithelium in the development of PanIN and PDAC and provide potential targets for therapeutic interventions.

### 3.2. The Role of IL-17/IL17R Families in Pancreatic Cancer

KRAS is one of the most common oncogenes in solid tumors. About 30% of tumors have KRAS mutations, including 90% of pancreatic cancers, 30–40% of colorectal cancers, and 15%-20% of lung cancers. KRAS G12D is the most common KRAS gene mutation in pancreatic cancer, accounting for about 41% [89]. KRAS-G12D expression promotes the infiltration of T cells expressing IL-17, and KRAS-G12D mutant pancreatic epithelial cells are induced by IL-17A, which activates the GP130-JAK2-STAT3 inflammatory pathway as a result of its stimulation [87]. As previously mentioned, IL-17 plays a crucial role in interacting with immune cells such as neutrophils and CD8+ T cells, as well as cytokines in the tumor microenvironment, contributing to the progression of various diseases, including pancreatic cancer (Figure 2D) [90]. Compared to healthy individuals, pancreatic cancer patients have remarkably higher serum levels of IL-17, and PDAC development and metastasis may be impacted by increasing levels of circulating Th17 cells and serum IL-17A [91,92]. Studies have shown that pancreatic cancer patients have a distinct tumor microenvironment, and those in stable or remission stages exhibit decreased levels of Treg cells, while IL-17A expression increases [10]. Conversely, patients with unresectable advanced pancreatic cancer show the opposite results, with increased levels of Treg cells and decreased IL-17A expression. These findings suggest that the balance between Treg cells and IL-17A expression may be a critical factor in the progression and prognosis of pancreatic cancer and could potentially be targeted for therapeutic interventions. Another study shows that Tc17, a new protumorigenic CD8+T-cell subtype in pancreatic cancer, enhanced tumor development via the IL-17A/RA pathway (Figure 2A) [93].

The IL-17B/RB pathway has been shown to enhance the expression of chemokines CCL20, CXCL1, IL-8, and TFF1 by regulating the ERK1/2 pathway [33], thereby promoting metastasis and malignancy in pancreatic cancer (Figure 2C). Furthermore, IL-17RB has been identified as a potential prognostic marker and a predictor of the effectiveness of gemcitabine treatment in patients with resectable pancreatic cancer [18]. Activating IL-17B/RB pathway in pancreatic stellate cells promotes pancreatic cancer metabolism and growth [94]. These findings suggest that the IL-17B/RB pathway may play a critical role in the progression and treatment of pancreatic cancer and represents a potential target for therapeutic interventions. Other studies have also explored the variant form of IL-17F and found it to be a proper antagonist of the anti-angiogenic effects of wild-type IL-17F, and angiogenesis may play a crucial role in the metastasis of pancreatic cancer [27]. Here, we have summarized these different ligands and receptors involved in the pathogenesis of pancreatic cancer in Table 2.

### 3.3. The Role of IL-17/IL17R Families in Chemotherapy Resistance

Chemotherapy is the most conventional therapy for pancreatic cancer, and gemcitabine is one of the first-line drugs used in treatment, providing satisfactory results [11]. However, clinical data has shown that many patients who receive chemotherapy show resistance within just a few weeks, and the efficacy of gemcitabine is limited by chemoresistance, which mainly arises from the activity of nucleoside transporters (NT) and nucleoside enzymes [97]. In addition, microenvironmental factors such as interleukins and TNF have been implicated in the development of chemoresistance [98]. Despite these challenges, efforts are underway to develop new strategies to overcome gemcitabine resistance and improve outcomes for patients with pancreatic cancer. IL-17 has been implicated in the development of chemotherapy resistance in a variety of cancer cell lines. Early studies have suggested that IL-17A can promote proliferation, migration, invasion, and resistance to chemotherapy-induced cell death in breast and colorectal cancer [99]. Meanwhile, in pancreatic cancer cells and tumor tissue, there is a positive correlation between the expression of IL-17RB and the expression of MUC1 and MUC4 [7]. When IL-17RB transcriptionally upregulates the expression of MUC1 and MUC4, cancer stem-like properties and resistance to gemcitabine are further enhanced [100]. These findings suggest that targeting IL-17RB and MUC4 may be a promising strategy to overcome gemcitabine resistance in pancreatic cancer. Some evidence has demonstrated that combining anti-IL-17A antibodies with gemcitabine represents an effective approach to enhancing the antitumor response by modulating macrophages, particularly in pancreatic cancer, for which gemcitabine is currently the most widely used anticancer drug [15]. These findings underscore the need for further research to better understand the mechanisms underlying IL-17-mediated chemoresistance and to develop new strategies for overcoming this challenge in cancer treatment.

## 4. Current Clinical Applications on Targeting IL-17/IL17R Family Member

IL-17 antibody drugs, including Secukinumab (Cosentyx), Brodalumab (Siliq), and Ixekizumab (Taltz), have been approved for the treatment of autoimmune diseases such as psoriasis [101]. Recent research has revealed that IL-17-producing cells are not limited to Th17 cells and has expanded the field of study beyond autoimmunity and inflammatory diseases to include tumor immunity [39]. As mentioned throughout this review, IL-17 can be involved in the regulation of tumor immunity and inflammation to produce protective or inhibitory effects on tumors. Additional IL-17-based immunotherapy may be possible with further characterization of immune profiles and elements influencing the functional plasticity of Th17 and T cells, even including the role of the microbiome [21]. To combat immunotherapy resistance, targeting Th17 cells’ functions may mitigate immune-related adverse effects without negatively impacting the antitumor efficacy of checkpoint inhibitors [102]. In a phase I clinical trial for multiple myeloma, patients are being examined to determine if anti-IL-17 mAb CJM112, either alone or in combination with anti-PD-1, is an effective treatment option to restore the immune response and favoring the apoptosis of tumor cells [103]. IL-17/IL-17R blockade remarkability enhanced sensitivity to checkpoint blockade therapy, and the combination of IL-17 therapies and checkpoint blockade PD-1, CTLA4 offers a unique treatment option that may be effective in treating this lethal disease by triggering neutrophil extracellular traps and excluding cytotoxic CD8^+^ T cells from pancreatic tumors [90]. There is some evidence that anti-IL-17 antibodies reduce the progression of pancreatic intraepithelial neoplasia and metastasis and enhance the effectiveness of anti-VEGF in colorectal cancer treatment [18,104]. IL-17A increases PD-L1 expression through the p65/NRF1/miR-15b-5p axis and promotes resistance to anti-PD-1 therapy, and IL-17F might be associated with a favorable overall survival in pancreatic cancer patients who were treated with gemcitabine [105,106]. The US FDA has approved several biologics for the treatment of autoinflammatory diseases by blocking IL-17 activity. However, the impact of IL-17 inhibitors on more common diseases remains unknown due to the absence of clinical trials. IL-17 family members may serve as biomarkers in predicting the prognosis of the tumor and the therapeutic benefits of immune checkpoint inhibitors, considering that recent research has examined IL-17 in pan-cancer and has discovered that the expression of IL-17 family is variable in different tumors [107]. Exploring this field could provide insight into how human cancer is affected by IL-17. In the future, therapeutic agents that block IL-17 activity may be used as boosters to overcome chemotherapy and radiotherapy resistance.

## 5. Future Perspectives

In recent years, a growing number of research in solid tumors have been found to exhibit IL-17/IL-17R-pathway and Th17-positive cells. Not only autoimmune diseases but also the members of the IL-17/IL-17R family have been revealed to influence the observed changes in cell proliferation, tumor growth and progression, and chemotherapy resistance. There has been much discussion over the last few years of IL-17 presenting in tumorigenesis by stimulating tumor angiogenesis and enhancing tumor immune evasion while exerting antitumor functions and recruiting immune cells into tumor tissues. We have discussed how the IL-17 family and their receptors with diseases and cancer in the level of tumor immunity and its role in inflammations. In particular, recent studies have shown that the IL-17 pathway has been a bridge to interact with different cancers such as breast, colorectal, and pancreatic cancer. In the treatment of pancreatic cancer, there is various new evidence investigating how IL-17 might be associated with the progression of pancreatic cancer and its TEM. Several studies have shown that pharmacotherapy targeting IL-17 might contribute to the therapeutic effect of pancreatic cancer if further investigation is carried out into various IL-17 family members and their receptors in pancreatic cancer pathogenesis. However, with the limitation of IL-17 research on pancreatic cancer, there is a clue showing that using these proximal IL-17 signaling mechanisms as a starting point would likely be adequate to block these key steps as a means of reducing the malignant potential of tumors, especially in the IL-17A/RA and IL-17B/RB axes. It may be necessary to develop biomarkers for localized IL-17 activity to identify patients who respond to clinical trials, and preclinical studies are still needed before a new treatment can be conducted. Overall, IL-17 represents a promising avenue for further investigation in the treatment of pancreatic cancer.

## Figures and Tables

**Figure 1 ijms-24-13539-f001:**
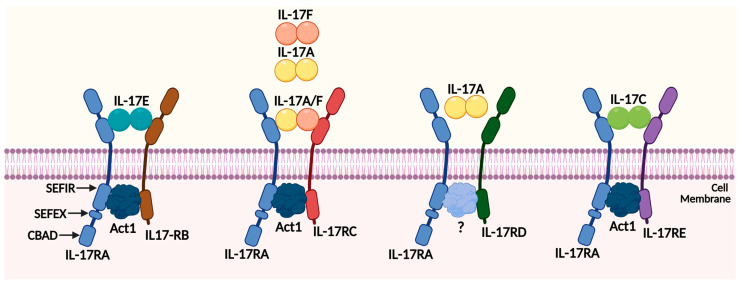
The cytokine ligands and receptors of the IL-17 family. The IL-17 family has six different cytokines’ ligands (IL-17A, IL-17B, IL-17C, IL-17D, IL-17E, and IL-17F), and IL-17A is the most prototypical cytokine, while five members of the IL-17 receptor (IL-17RA, IL-17RB, IL-17RC, IL-17RD, IL-17RE) have been found in previous publications. IL-17RA is the most common receptor to form heterodimeric complexes with other subunits. The IL-17RA/IL-17RB complex binds the IL-17E ligand, and the IL-17A and IL-17F are individually or concurrently received by IL-17RA/IL-17RC and IL-17RA/IL-17RD. IL-17RA/IL-17RD recruits an unknown protein through SEFIR-SEFIR domain interaction. IL-17RA/IL-17RE bind IL-17C. Abbreviations: CBAD, C/EBPb activation domain; SEFEX, SEFIR extension; SEFIR, similar expression of fibroblast growth factor and IL-17Rs.

**Figure 2 ijms-24-13539-f002:**
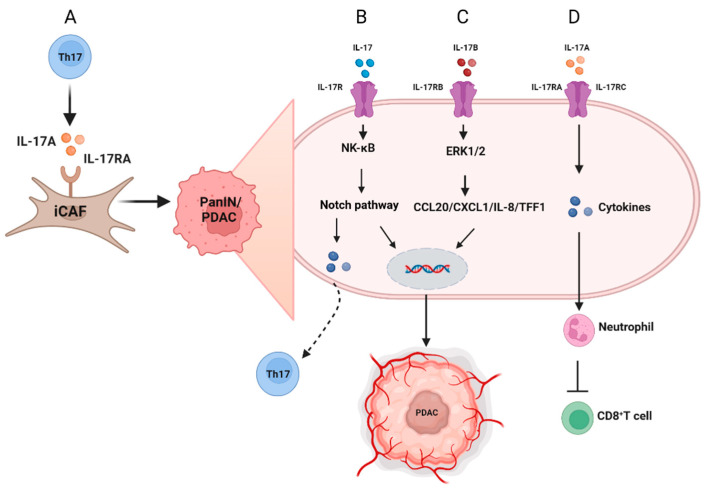
Schematic diagram of the role of IL-17 in the pathogenesis and intracellular pathway of pancreatic cancer. (**A**) Th17 cells can promote iCAF formation via IL-17A secretion and are further involved in the progression of pancreatic cancer. (**B**) IL-17 and Notch are overexpressed in PanIN and PDAC while promoting the progression of PanIN/PDAC mediated by the NF-κB pathway, and the Notch pathway is pivotal in the differentiation of the Th17 cells (black dashed line). (**C**) The IL-17B–IL-17RB axis activated CCL20/CXCL1/IL-8/TFF1 chemokine expressions via the ERK1/2 pathway to promote metastasis of pancreatic cancer cells. (**D**) IL-17A bound IL-17RA/RC complexes to secrete cytokines to recruit neutrophiles, which inhibited CD8^+^ T cells in a pancreatic tumor microenvironment.

**Table 1 ijms-24-13539-t001:** Summary of IL-17 ligands and receptors.

Cytokine Ligands	Receptors	Relative Cell Types	Sources	Downstream Functions
IL-17A	IL-17RA and IL-17RC IL-17RA and IL-17RD	Th17 cells, αβ T cells, γδ T cells, iNKT cells, and LTi-like cells	Skin, gut, lung, pancreas	Proinflammatory; Promote cancer progression [13]; Regulate immune cells; T cell activation to neutrophil mobilization and activation [42]
IL-17B	IL-17RB	Neutrophils, chondrocytes, neurons, naïve, memory, and germinal center B cells	Stomach, pancreas, intestine, rheumatoid synovial tissues	Control immune cell trafficking to inflamed tissues; Promote cancer cell survival, proliferation and migration [16];
IL-17C	IL-17RA and IL-17RE	Epithelial cells	In the mucosa of the mouth, skin, airway epithelium thymus and spleen	Regulate the innate immune function of epithelial cells [43]; Induce innate immune functions in bacterial, fungal, and brain infections [44]; Host defense against pathogens [45]
IL-17D	Unknown	T cells, smooth muscle cells, epithelial cells, mast cells	Skeletal muscle, brain, adipose tissue, heart, lung and pancreas [23]	Against dextran sulfate sodium (DSS)-induced colitis [46]
IL-17E	IL-17RA and IL-17RB	Mast cells, Epithelial cells, Th2 cells, NKT cells, Mast cells	Brain, kidney, lung, prostate, testis, spinal cord, adrenal gland	Anti-inflammatory and proinflammatory; Promote both proliferation and differentiation of keratinocytes [47]
IL-17F	IL-17RA and IL-17RC	Th17 cells, neutrophil, NK cells, γδ T cells, Mast cells	Skin, Joint	Antitumor effects; Neutrophil recruitments; Regulate the expression of inflammatory chemokines and cytokines [48]

**Table 2 ijms-24-13539-t002:** IL-17 family in pathogenesis of pancreatic cancer.

IL-17 Subtype and Receptors	Mechanism Models	Biological Effects	Reference
IL-17A	Il-17A expression increased	Patients with PC in stable and remission stages	[95]
	Il-17A expression decreased	Patients with unresectable advanced PC	
	IL-17 exerts effects on IL-17RA of PanIN epithelial cells	Accelerating the progression and PanIN	[84]
	IL-17A recruits neutrophils	Immunosuppressive microenvironment in PDAC	[90]
	Il-17 A-producing CD8+ T cells	Promoting PDAC progression	[70,93]
	IL-17 A activates the gp130-JAK2-STAT3 pathway	Promoting acinar-ductal metaplasia and PanIN development	[87]
IL-17B/IL-17RB	Activating the ERK1/2 signaling pathway	Promoting the invasion and metastasis of PC	[18]
	IL-17RB expression increased	Predicting the prognosis of patients with resectable PC	[96]
	Anti-IL-17RB monoclonal antibody	Inhibiting of tumor metastasis	[16]
IL-17E	Activating the NF-κB signaling pathway	Exerting an antitumor effect in combination with drugs	[25]

## Data Availability

Not applicable.

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
