# Peer review of "Mechanistic Insights into the Roles of the IL-17/IL-17R Families in Pancreatic Cancer"

_ijms, 2023, doi:10.3390/ijms241713539_

Round 1
Reviewer 1 Report
The review by Chen t al., describes potential role of IL-17/IL17R families in pancreatic cancer. In general it is an interesting and promising topic. The manuscript provides a lot of interesting information; however, there are a few issues the authors should amend. I believe there are several strongly formulated statements without a solid scientific ground.
Specific comments:
1. Line 45. I think the authors should either modify the sentence or explain with references what they mean by “The tumor environment maintained by Th17…”. What do you mean by maintained?
2. Line 81. How would the combined treatment regulate M1 and M2? What do you mean by “regulate”?
3. A different paragraph should be made for each IL described in the section 2.1.
4. I find section 2.4. misleading as it does really describe the role of IL in inflammation but rather implicates IL in tumorigenesis and the disease progression with some impact on inflammation in tumour.
5. The same applies to the next section. There is very little about tumour immunity but rather about tumour progression. Maybe it would make more sense to combine the two sections and discuss the impact of IL-17 on tumour progression indicating the importance of inflammation and immunity.
6. The statement on pancreatic cancer being a devastating disease is often repeated in the ms. Please remove where not necessary.
7. Line 311. Please correct the sentence.
8. Line 363. The statement about IL-17-mediated resistance is very strong and I could not really see any proper justification in the text. You either need to give scientific justification with references or soften the statement.
N/A
Author Response
Dear Reviewer:
Thank you for the critical comments and helpful suggestions. We have taken all these comments and suggestions into account, and have made major corrections in this revised manuscript:
Comment 1: Line 45. I think the authors should either modify the sentence or explain with references what they mean by “The tumor environment maintained by Th17…”. What do you mean by maintained?
Response: Thank you for the comment, we wanted to discuss the Th17 and other immune cells modulate the tumour microenvironment by interacting in various interleukins or other IL-17 cytokines. We think it’s better to say the tumour microenvironment is regulated by Th17 cells and other immune cells since we mentioned that the proportion of these cells caused different dynamics in tumour immunity in this review. In Line 49-51, we have modified this sentence and cited a new reference.
Comment 2: Line 81. How would the combined treatment regulate M1 and M2? What do you mean by “regulate”?
Response: We have clarified the sentence in Line 86, it is better to mention that combination treatment regulates the function and activity such as the metabolism of macrophages in TME, specifically, decreases the M1 and M2 metabolic pathways.
Comment 3: A different paragraph should be made for each IL described in the section 2.1.
Response: We think that is a brilliant suggestion, we have already separated several individual paragraphs, and the whole structure looks clearer and more organized. Thank you
Comment 4: I find section 2.4. misleading as it does really describe the role of IL in inflammation but rather implicates IL in tumorigenesis and the disease progression with some impact on inflammation in tumour. The same applies to the next section. There is very little about tumour immunity but rather about tumour progression. Maybe it would make more sense to combine the two sections and discuss the impact of IL-17 on tumour progression indicating the importance of inflammation and immunity.
Response: Thank you for your excellent suggestions, we have combined two sections and also discussed more details of IL-17 on inflammation and tumour immunity.
Comment 5 The statement on pancreatic cancer being a devastating disease is often repeated in the ms. Please remove where not necessary
Response: Thank you for your suggestions, we have removed and revised sort of this statement.
Comment 6 Line 311. Please correct the sentence.
Response: We are very sorry for our incorrect writing, and we have tried to polish this sentence in the revised manuscript in the Line 365.
Comment 7: Line 363. The statement about IL-17-mediated resistance is very strong and I could not really see any proper justification in the text. You either need to give scientific justification with references or soften the statement.
Response: It is really true as the reviewer suggested that IL-17-mediated resistance is very strong, and we have modified this sentence in Line 427
Reviewer 2 Report
The review highlights the important role of significant role of the cytokine interleukin 17 (IL-17) family and their receptors (IL-17R) in both inflammatory diseases and cancer. Also, authors tried to emphasize the potential of IL-17/IL-17R as not only prognostic markers but also as therapeutic targets for pancreatic cancer. It is an interesting review and certainly add value to the readers, however important consideration must be made to improve the article:
1. Few typos were observed, authors must check it carefully
2. A table must be made showing different ligands and receptors involved in pancreatic cancer pathogenesis with a column highlighting mechanism of action
3. In "Current Clinical Applications on targeting IL-17/IL17R family member", the authors mentioned about IL17 targeting drugs, however how mechanism associated with drugs are not discussed. Please give details.
4. A dedicated section on the influence of IL17 on immunotherapy must be discussed as it is an important molecule in dictating immunotherapeutic response in different cancers including pancreatic cancer.
Author Response
Dear reviewer,
We feel great thanks for your professional comments and suggestions on our review. As you are concerned, there are several problems that need to be addressed. and we were really sorry for our careless mistakes, thank you for your reminder. And We also have made extensive corrections to our previous draft.
Comment 1 Few typos were observed, authors must check it carefully
Response: We are very sorry for our negligence in writing. In our resubmitted manuscript, the typos are revised. Thank you for your correction
Comment 2: A table must be made showing different ligands and receptors involved in pancreatic cancer pathogenesis with a column highlighting the mechanism of action
Response: We have made a new table for showing the function of ligands and receptors in pancreatic cancer
Comment 3: In "Current Clinical Applications on targeting IL-17/IL17R family member", the authors mentioned about IL17 targeting drugs, however how mechanism associated with drugs are not discussed. Please give details
Response: We have added more references and details in section 4 and explained most of the mechanism and signalling behind these drugs or immunotherapy
Comment 4: A dedicated section on the influence of IL17 on immunotherapy must be discussed as it is an important molecule in dictating immunotherapeutic response in different cancers including pancreatic cancer
Response: We think this is an excellent suggestion. We have added more statements of IL717 on immunotherapy in section 4
Reviewer 3 Report
The review article titled "Mechanistic Insights into the Roles of the IL-17/IL-17R families in Pancreatic Cancer" aimed to discuss the role of IL-17/IL-17R in PAD and PDAC and chemotherapy resistance. However, after abstract and introduction, the review is written in a very naive way. Section 2.1 should be briefly discussed using a schematic as included.
Section 2.2 should be summarized using a table and figure.
sections 2.3 and 2.4 should be discussed in the perspective of PDAC and not general description.
Similarly, section 2.5 should be discussed in relation to PDAC.
section 3.1- there is no need to discuss the pathogenesis of PDAC, it is available in the literature. The authors should focus the pathogenesis in the perspective of IL-17. There is enough literature to include. Like https://gut.bmj.com/content/72/8/1510;https://pubmed.ncbi.nlm.nih.gov/36759154/;https://www.ncbi.nlm.nih.gov/pmc/articles/PMC4259416/;https://rupress.org/jem/article/217/12/e20190354/152058/Interleukin-17-induced-neutrophil-extracellular;https://rupress.org/jem/article/217/1/e20190297/132590/The-role-of-interleukin-17-in-tumor-development;https://www.nature.com/articles/s41575-023-00757-4;https://www.nature.com/articles/nrd4372;https://www.cell.com/cancer-cell/pdfExtended/S1535-6108(14)00122-6;https://www.frontiersin.org/articles/10.3389/fimmu.2022.900273/full;https://www.mendeley.com/catalogue/5d1718dd-a60f-346a-b47f-a491c50bf6fb/;https://www.sciencedirect.com/science/article/pii/S0304383521001075;https://www.pnas.org/doi/10.1073/pnas.2020395118;https://www.researchgate.net/publication/262303691_Oncogenic_Kras_Activates_a_Hematopoietic-to-Epithelial_IL-17_Signaling_Axis_in_Preinvasive_Pancreatic_Neoplasia to name a few. None of these aspects or only a few has been included in relation to PDAC and IL-17.
Please elaborate section 3 and 4.
Author Response
Dear reviewer,
I would like to express our sincere appreciation for your constructive comments. These comments and suggestions are all valuable and helpful for improving our review. We have made extensive modifications to our manuscript:
Comment 1: Section 2.1 should be briefly discussed using a schematic as included.
Response: Since we used Figure 1 to summarise IL-17 cytokines and coupled receptors, we have added more contents and references to discuss this figure in Line 184-196, especially in the molecular level of IL-17, which might get more clarified in section 2.1. Thank you for your constructive suggestions.
Comment 2:Section 2.2 should be summarized using a table and figure.
Response: We have added a table to summarize all the ligands and coupled receptors in section 2.2, and the IL-17 relative cell types and its function and mechanisms have been included
Comment 3:sections 2.3 and 2.4 should be discussed from the perspective of PDAC and not a general description. sections 2.3 and 2.4 should be discussed from the perspective of PDAC and not a general description.
Response: Thank you for your comments, we actually have summarized most of the perspective and discussion of PDAC in section 3 as an individual part. In sections 2.3/2.4 we were thinking of discussing the whole scope of IL-17 since IL-17 also plays an important role and has many functions in different cancer types and diseases.
Comment 4:section 3.1- there is no need to discuss the pathogenesis of PDAC, it is available in the literature. The authors should focus on the pathogenesis in the perspective of IL-17. Please elaborate section 3 and 4.
Response: Thank you for all this significant literature, we have checked and discussed all these references and added it all in sections 3 and 4.
Reviewer 4 Report
This review deals with the relevance of IL17 in pancreatic cancer. First, the authors describe the feature of IL17 family and their receptor with some observation on pancreatic cancer. Afterwards, they describe these components in pancreatic cancer and finally, they considered the possibility of targeting IL17-IL17R signalling to regulate the biology of pancreatic tumor cells.
The manuscript is focused on pancreatic cells, and I would imagine that a review similar can be set up for several other kinds of cancers. The review itself can give some interesting suggestions and references on the topic. It is to determine if this work is publishable, considering that a similar review ca be prepared for many other kinds of cancers and of course of cytokines.
If accepted by the other reviewers, the authors should add the reference below and discuss it
Khan IA, Singh N, Gunjan D, Gopi S, Dash NR, Gupta S, Saraya A. Increased circulating Th17 cell populations in patients with pancreatic ductal adenocarcinoma.Immunogenetics. 2023 Aug 4. doi: 10.1007/s00251-023-01318-4. Online ahead of print.
The English language is good.
Author Response
Dear reviewer,
We would like to thank you for your careful reading, helpful comments, and constructive suggestions, which have significantly improved the presentation of our manuscript. We hope our revised manuscript can be accepted for publication
Comment 1:If accepted by the other reviewers, the authors should add the reference below and discuss it
Khan IA, Singh N, Gunjan D, Gopi S, Dash NR, Gupta S, Saraya A. Increased circulating Th17 cell populations in patients with pancreatic ductal adenocarcinoma. Immunogenetics. 2023 Aug 4. doi: 10.1007/s00251-023-01318-4. Online ahead of print.
Response: We sincerely appreciate the valuable comments, we have checked this latest reference carefully and added it to our review in line 364-367 (Please see the attachment). We think it is very good literature to understand circulating Th17 cells in pancreatic cancer patients. And we think it might offer a new thought or strategy for clinical treatment.

Round 2
Reviewer 3 Report
none
Reviewer 4 Report
The authors following the reviewers comment have improved the manuscript. I think it can be a good starting point to focus on the IL17 in pancreatic cancer.